# Robust Hypothesis Test for Nonlinear Effect with Gaussian Processes

**Jeremiah Zhe Liu, Brent Coull**
Department of Biostatistics
Harvard University
Cambridge, MA 02138
{zhl112@mail, bcoull@hsph}.harvard.edu

## Abstract

This work constructs a hypothesis test for detecting whether an data-generating function $h : \mathbb{R}^p \to \mathbb{R}$ belongs to a specific reproducing kernel Hilbert space $\mathcal{H}_0$, where the structure of $\mathcal{H}_0$ is only partially known. Utilizing the theory of reproducing kernels, we reduce this hypothesis to a simple one-sided score test for a scalar parameter, develop a testing procedure that is robust against the mis-specification of kernel functions, and also propose an ensemble-based estimator for the null model to guarantee test performance in small samples. To demonstrate the utility of the proposed method, we apply our test to the problem of detecting nonlinear interaction between groups of continuous features. We evaluate the finite-sample performance of our test under different data-generating functions and estimation strategies for the null model. Our results reveal interesting connections between notions in machine learning (model underfit/overfit) and those in statistical inference (i.e. Type I error/power of hypothesis test), and also highlight unexpected consequences of common model estimating strategies (e.g. estimating kernel hyperparameters using maximum likelihood estimation) on model inference.

## 1 Introduction

We study the problem of constructing a hypothesis test for an unknown data-generating function $h : \mathbb{R}^p \to \mathbb{R}$, when $h$ is estimated with a black-box algorithm (specifically, Gaussian Process regression) from $n$ observations $\{y_i, \mathbf{x}_i\}_{i=1}^n$. Specifically, we are interested in testing for the hypothesis:

$$H_0 : h \in \mathcal{H}_0 \qquad v.s. \qquad H_a : h \in \mathcal{H}_a$$

where $\mathcal{H}_0, \mathcal{H}_a$ are the function spaces for $h$ under the null and the alternative hypothesis. We assume only partial knowledge about $\mathcal{H}_0$. For example, we may assume $\mathcal{H}_0 = \{h | h(\mathbf{x}_i) = h(x_{i,1})\}$ is the space of functions depend only on $x_1$, while claiming no knowledge about other properties (linearity, smoothness, etc) about $h$. We pay special attention to the setting where the sample size $n$ is small.

This type of tests carries concrete significance in scientific studies. In areas such as genetics, drug trials and environmental health, a hypothesis test for feature effect plays a critical role in answering scientific questions of interest. For example, assuming for simplicity $\mathbf{x}_{2 \times 1} = [x_1, x_2]^T$, an investigator might inquire the effect of drug dosage $x_1$ on patient's biometric measurement $y$ (corresponds to the null hypothesis $\mathcal{H}_0 = \{h(\mathbf{x}) = h(x_2)\}$), or whether the adverse health effect of air pollutants $x_1$ is modified by patients' nutrient intake $x_2$ (corresponds to the null hypothesis $\mathcal{H}_0 = \{h(\mathbf{x}) = h_1(x_1) + h_2(x_2)\}$). In these studies, $h$ usually represents some complex, nonlinear biological process whose exact mathematical properties are not known. Sample size in these studies are often small ($n \approx 100 - 200$), due to the high monetary and time cost in subject recruitment and the lab analysis of biological samples.

There exist two challenges in designing such a test. The first challenge arises from the low interpretability of the blackbox model. It is difficult to formulate a hypothesis about feature effect in these models, since the blackbox models represents $\hat{h}$ implicitly using a collection of basis functions constructed from the entire feature vector $\mathbf{x}$, rather than a set of model parameters that can be interpreted in the context of some effect of interest. For example, consider testing for the interaction effect between $x_1$ and $x_2$. With linear model $h(\mathbf{x}_i) = x_{i1}\beta_1 + x_{i2}\beta_2 + x_{i1}x_{i2}\beta_3$, we can simply represent the interaction effect using a single parameter $\beta_3$, and test for $H_0 : \beta_3 = 0$. On the other hand, Gaussian process (GP) [16] models $h(\mathbf{x}_i) = \sum_{j=1}^{n} k(\mathbf{x}_i, \mathbf{x}_j)\alpha_j$ using basis functions defined by the kernel function $k$. Since $k$ is an implicit function that takes the entire feature vector as input, it is not clear how to represent the interaction effect in GP models. We address this challenge assuming $h$ belongs to a reproducing kernel Hilbert space (RKHS) governed by the kernel function $k_\delta$, such that $\mathcal{H} = \mathcal{H}_0$ when $\delta = 0$, and $\mathcal{H} = \mathcal{H}_a$ otherwise. In this way, $\delta$ encode exactly the feature effect of interest, and the null hypothesis $h \in \mathcal{H}_0$ can be equivalently stated as $H_0 : \delta = 0$. To test for the hypothesis, we re-formulate the GP estimates as the variance components of a linear mixed model (LMM) [13], and derive a variance component score test which requires only model estimates under the null hypothesis.

Clearly, performance of the hypothesis test depends on the quality of the model estimate under the null hypothesis, which give rise to the second challenge: estimating the null model when only having partial knowledge about $\mathcal{H}_0$. In the case of Gaussian process, this translates to only having partial knowledge about the kernel function $k_0$. The performance of Gaussian process is sensitive to the choices of the kernel function $k(\mathbf{z}, \mathbf{z}')$. In principle, the RKHS $\mathcal{H}$ generated by a proper kernel function $k(\mathbf{z}, \mathbf{z}')$ should be rich enough so it contains the data-generating function $h$, yet restrictive enough such that $\hat{h}$ does not overfit in small samples. Choosing a kernel function that is too restrictive or too flexible will lead to either model **underfit** or **overfit**, rendering the subsequent hypothesis tests not valid. We address this challenge by proposing an ensemble-based estimator for $h$ we term *Cross-validated Kernel Ensemble* (CVEK). Using a library of base kernels, CVEK learns a proper $\mathcal{H}$ from data by directly minimizing the ensemble model's cross-validation error, therefore guaranteeing robust test performance for a wide range of data-generating functions.

The rest of the paper is structured as follows. After a brief review of Gaussian process and its connection with linear mixed model in Section 2, we introduce the test procedure for general hypothesis $h \in \mathcal{H}_0$ in Section 3, and its companion estimation procedure CVEK in Section 4. To demonstrate the utility of the proposed test, in section 5, we adapt our test to the problem of detecting nonlinear interaction between groups of continuous features, and in section 6 we conduct simulation studies to evaluate the finite-sample performance of the interaction test, under different kernel estimation strategies, and under a range of data-generating functions with different mathematical properties. Our simulation study reveals interesting connection between notions in machine learning and those in statistical inference, by elucidating the consequence of model estimation (underfit / overfit) on the Type I error and power of the subsequent hypothesis test. It also cautions against the use of some common estimation strategies (most notably, selecting kernel hyperparameters using maximum likelihood estimation) when conducting hypothesis test in small samples, by highlighting inflated Type I errors from hypothesis tests based on the resulting estimates. We note that the methods and conclusions from this work is extendable beyond the Gaussian Process models, due to GP's connection to other blackbox models such as random forest [5] and deep neural network [19].

## 2 Background on Gaussian Process

Assume we observe data from $n$ independent subjects. For the $i^{th}$ subject, let $y_i$ be a continuous response, $\mathbf{x}_i$ be the set of $p$ continuous features that has nonlinear effect on $y_i$. We assume that the outcome $y_i$ depends on features $\mathbf{x}_i$ through below data-generating model:

$$y_i | h = \mu + h(\mathbf{x}_i) + \epsilon_i \qquad \text{where } \epsilon_i \overset{iid}{\sim} N(0, \lambda) \tag{1}$$

and $h : \mathbb{R}^p \to \mathbb{R}$ follows the Gaussian process prior $\mathcal{GP}(0, k)$ governed by the positive definite kernel function $k$, such that the function evaluated at the observed record follows the multivariate normal (MVN) distribution:

$$\mathbf{h} = [h(\mathbf{x}_1), \dots, h(\mathbf{x}_n)] \sim MVN(\mathbf{0}, \mathbf{K})$$

with covariance matrix $\mathbf{K}_{i,j} = k(\mathbf{x}_i, \mathbf{x}_j)$. Under above construction, the predictive distribution of $h$ evaluated at the samples is also multivariate normal:

$$\mathbf{h}|\{y_i, \mathbf{x}_i\}_{i=1}^n \sim MVN(\mathbf{h}^*, \mathbf{K}^*)$$
$$\mathbf{h}^* = \mathbf{K}(\mathbf{K} + \lambda\mathbf{I})^{-1}(\mathbf{y} - \mathbf{u})$$
$$\mathbf{K}^* = \mathbf{K} - \mathbf{K}(\mathbf{K} + \lambda\mathbf{I})^{-1}\mathbf{K}$$

To understand the impact of $\lambda$ and $k$ on $\mathbf{h}^*$, recall that Gaussian process can be understood as the Bayesian version of the kernel machine regression, where $\mathbf{h}^*$ equivalently arise from the below optimization problem:

$$\mathbf{h}^* = \underset{h \in \mathcal{H}_k}{argmin} \, ||\mathbf{y} - \boldsymbol{\mu} - h(\mathbf{x})||^2 + \lambda||h||_{\mathcal{H}}^2$$

where $\mathcal{H}_k$ is the RKHS generated by kernel function $k$. From this perspective, $\mathbf{h}^*$ is the element in a spherical ball in $\mathcal{H}_k$ that best approximates the observed data $\mathbf{y}$. The norm of $\mathbf{h}^*$, $||h||_{\mathcal{H}}^2$, is constrained by the tuning parameter $\lambda$, and the mathematical properties (e.g. smoothness, spectral density, etc) of $\mathbf{h}^*$ are governed by the kernel function $k$. It should be noticed that although $\mathbf{h}^*$ may arise from $\mathcal{H}_k$, the probability of the Gaussian Process $\mathbf{h} \in \mathcal{H}_k$ is 0 [14].

**Gaussian Process as Linear Mixed Model**

[13] argued that if define $\tau = \frac{\sigma^2}{\lambda}$, $\mathbf{h}^*$ can arise exactly from a linear mixed model (LMM):

$$\mathbf{y} = \boldsymbol{\mu} + \mathbf{h} + \boldsymbol{\epsilon} \qquad \text{where} \qquad \mathbf{h} \sim N(\mathbf{0}, \tau\mathbf{K}) \qquad \boldsymbol{\epsilon} \sim N(\mathbf{0}, \sigma^2\mathbf{I}) \qquad (2)$$

Therefore $\lambda$ can be treated as part of the LMM's variance components parameters. If $\mathbf{K}$ is correctly specified, then the variance components parameters $(\tau, \sigma^2)$ can be estimated unbiasedly by maximizing the Restricted Maximum Likelihood (REML)[12]:

$$L_{\text{REML}}(\boldsymbol{\mu}, \tau, \sigma^2|\mathbf{K}) = -log|\mathbf{V}| - log|\mathbf{1}^T\mathbf{V}^{-1}\mathbf{1}| - (\mathbf{y} - \boldsymbol{\mu})^T\mathbf{V}^{-1}(\mathbf{y} - \boldsymbol{\mu}) \qquad (3)$$

where $\mathbf{V} = \tau\mathbf{K} + \sigma^2\mathbf{I}$, and $\mathbf{1}$ a $n \times 1$ vector whose all elements are 1. However, it is worth noting that REML is a model-based procedure. Therefore improper estimates for $\lambda = \frac{\sigma^2}{\tau}$ may arise when the family of kernel functions are mis-specified.

# 3 A recipe for general hypothesis $h \in \mathcal{H}_0$

The GP-LMM connection introduced in Section 2 opens up the arsenal of statistical tools from Linear Mixed Model for inference tasks in Gaussian Process. Here, we use the classical variance component test [12] to construct a testing procedure for the hypothesis about Gaussian process function:

$$H_0 : h \in \mathcal{H}_0. \qquad (4)$$

We first translate above hypothesis into a hypothesis in terms of model parameters. The key of our approach is to assume that $h$ lies in a RKHS generated by a *garrote kernel function* $k_\delta(\mathbf{z}, \mathbf{z}')$ [15], which is constructed by including an extra *garrote parameter* $\delta$ to a given kernel function. When $\delta = 0$, the garrote kernel function $k_0(\mathbf{x}, \mathbf{x}') = k_\delta(\mathbf{x}, \mathbf{x}')\big|_{\delta=0}$ generates exactly $\mathcal{H}_0$, the space of functions under the null hypothesis. In order to adapt this general hypothesisio to their hypothesis of interest, practitioners need only to specify the form of the garrote kernel so that $\mathcal{H}_0$ corresponds to the null hypothesis. For example, If $k_\delta(\mathbf{x}) = k(\delta * x_1, x_2, \ldots, x_p)$, $\delta = 0$ corresponds to the null hypothesis $H_0 : h(\mathbf{x}) = h(x_2, \ldots, x_p)$, i.e. the function $h(\mathbf{x})$ does not depend on $x_1$. (As we'll see in section 5, identifying such $k_0$ is not always straightforward). As a result, the general hypothesis (4) is equivalent to

$$H_0 : \delta = 0. \qquad (5)$$

We now construct a test statistic $\hat{T}_0$ for (5) by noticing that the garrote parameter $\delta$ can be treated as a variance component parameter in the linear mixed model. This is because the Gaussian process under garrote kernel can be formulated into below LMM:

$$\mathbf{y} = \boldsymbol{\mu} + \mathbf{h} + \boldsymbol{\epsilon} \qquad \text{where} \qquad \mathbf{h} \sim N(\mathbf{0}, \tau\mathbf{K}_\delta) \qquad \boldsymbol{\epsilon} \sim N(\mathbf{0}, \sigma^2\mathbf{I})$$

where $\mathbf{K}_\delta$ is the kernel matrix generated by $k_\delta(\mathbf{z}, \mathbf{z}')$. Consequently, we can derive a variance component test for $H_0$ by calculating the square derivative of $L_{\text{REML}}$ with respect $\delta$ under $H_0$ [12]:

$$\hat{T}_0 = \hat{\tau} * (\mathbf{y} - \hat{\boldsymbol{\mu}})^T \mathbf{V}_0^{-1} \Big[\partial \mathbf{K}_0\Big] \mathbf{V}_0^{-1}(\mathbf{y} - \hat{\boldsymbol{\mu}}) \tag{6}$$

where $\mathbf{V}_0 = \hat{\sigma}^2 \mathbf{I} + \hat{\tau}\mathbf{K}_0$. In this expression, $\mathbf{K}_0 = \mathbf{K}_\delta\Big|_{\delta=0}$, and $\partial \mathbf{K}_0$ is the null derivative kernel matrix whose $(i,j)^{th}$ entry is $\frac{\partial}{\partial \delta} k_\delta(\mathbf{x}, \mathbf{x}')\Big|_{\delta=0}$.

As discussed previously, misspecifying the null kernel function $k_0$ negatively impacts the performance of the resulting hypothesis test. To better understand the mechanism at play, we express the test statistic $\hat{T}_0$ from (6) in terms of the model residual $\hat{\boldsymbol{\epsilon}} = \mathbf{y} - \hat{\boldsymbol{\mu}} - \hat{\mathbf{h}}$:

$$\hat{T}_0 = \left(\frac{\hat{\tau}}{\hat{\sigma}^4}\right) * \hat{\boldsymbol{\epsilon}}^T \Big[\partial \mathbf{K}_0\Big]\hat{\boldsymbol{\epsilon}}, \tag{7}$$

where we have used the fact $\mathbf{V}_0^{-1}(\mathbf{y} - \hat{\boldsymbol{\mu}}) = (\hat{\sigma}^2)^{-1}(\hat{\boldsymbol{\epsilon}})$ [10]. As shown, the test statistic $\hat{T}_0$ is a scaled quadratic-form statistic that is a function of the model residual. If $k_0$ is too restrictive, model estimates will **underfit** the data even under the null hypothesis, introducing extraneous correlation among the $\hat{\epsilon}_i$'s, therefore leading to overestimated $\hat{T}_0$ and eventually underestimated p-value under the null. In this case, the test procedure will frequently reject the null hypothesis (i.e. suggest the existence of nonlinear interaction) even when there is in fact no interaction, yielding an invalid test due to **inflated Type I error**. On the other hand, if $k_0$ is too flexible, model estimates will likely **overfit** the data in small samples, producing underestimated residuals, an underestimated test statistic, and overestimated p-values. In this case, the test procedure will too frequently fail to reject the null hypothesis (i.e. suggesting there is no interaction) when there is in fact interaction, yielding an insensitive test with **diminished power**.

The null distribution of $\hat{T}$ can be approximated using a scaled chi-square distribution $\kappa \chi_\nu^2$ using Satterthwaite method [20] by matching the first two moments of $T$:

$$\kappa * \nu = E(T), \qquad 2 * \kappa^2 * \nu = Var(T)$$

with solution (see Appendix for derivation):

$$\hat{\kappa} = \hat{\mathbf{I}}_{\delta\delta}/\Big[\hat{\tau} * tr\Big(\mathbf{V}_0^{-1}\partial \mathbf{K}_0\Big)\Big] \qquad \hat{\nu} = \Big[\hat{\tau} * tr\Big(\mathbf{V}_0^{-1}\partial \mathbf{K}_0\Big)\Big]^2/(2 * \hat{\mathbf{I}}_{\delta\delta})$$

where $\hat{\mathbf{I}}_{\delta\theta}$ and $\hat{\mathbf{I}}_{\delta\theta}$ are the submatrices of the REML information matrix. Numerically more accurate, but computationally less efficient approximation methods are also available [2].

Finally, the p-value of this test is calculated by examining the tail probability of $\hat{\kappa}\chi_\nu^2$:

$$p = P(\hat{\kappa}\chi_{\hat{\nu}}^2 > \hat{T}) = P(\chi_{\hat{\nu}}^2 > \hat{T}/\hat{\kappa})$$

A complete summary of the proposed testing procedure is available in Algorithm 1.

---
**Algorithm 1** Variance Component Test for $h \in \mathcal{H}_0$
---
1: **procedure** VCT FOR INTERACTION
    **Input:** Null Kernel Matrix $\mathbf{K}_0$, Derivative Kernel Matrix $\partial \mathbf{K}_0$, Data $(\mathbf{y}, \mathbf{X})$
    **Output:** Hypothesis Test p-value $p$
    # Step 1: Estimate Null Model using REML
2:    $(\hat{\boldsymbol{\mu}}, \hat{\tau}, \hat{\sigma}^2) = argmin\ L_{\text{REML}}(\boldsymbol{\mu}, \tau, \sigma^2 | \mathbf{K}_0)$ as in (3)
    # Step 2: Compute Test Statistic and Null Distribution Parameters
3:    $\hat{T}_0 = \hat{\tau} * (\mathbf{y} - \mathbf{X}\hat{\boldsymbol{\beta}})^T \mathbf{V}_0^{-1}\ \partial \mathbf{K}_0\ \mathbf{V}_0^{-1}(\mathbf{y} - \mathbf{X}\hat{\boldsymbol{\beta}})$
4:    $\hat{\kappa} = \hat{\mathbf{I}}_{\delta\theta}/\Big[\hat{\tau} * tr\Big(\mathbf{V}_0^{-1}\partial \mathbf{K}_0\Big)\Big], \quad \hat{\nu} = \Big[\hat{\tau} * tr\Big(\mathbf{V}_0^{-1}\partial \mathbf{K}_0\Big)\Big]^2/(2 * \hat{\mathbf{I}}_{\delta\theta})$
    # Step 3: Compute p-value and reach conclusion
5:    $p = P(\hat{\kappa}\chi_{\hat{\nu}}^2 > \hat{T}) = P(\chi_{\hat{\nu}}^2 > \hat{T}/\hat{\kappa})$
6: **end procedure**
---

In light of the discussion about model misspecification in Introduction section, we highlight the fact that our proposed test (6) is robust against model misspecification under the alternative [12], since the calculation of test statistics do not require detailed parametric assumption about $k_\delta$. However, the test is NOT robust against model misspecification under the null, since the expression of both test statistic $\hat{T}_0$ and the null distribution parameters $(\hat{\kappa}, \hat{\nu})$ still involve the kernel matrices generated by $k_0$ (see Algorithm 1). In the next section, we address this problem by proposing a robust estimation procedure for the kernel matrices under the null.

## 4 Estimating Null Kernel Matrix using Cross-validated Kernel Ensemble

Observation in (7) motivates the need for a kernel estimation strategy that is *flexible* so that it does not underfit under the null, yet *stable* so that it does not overfit under the alternative. To this end, we propose estimating $h$ using the ensemble of a library of fixed base kernels $\{k_d\}_{d=1}^D$:

$$\hat{h}(\mathbf{x}) = \sum_{d=1}^D u_d \hat{h}_d(\mathbf{x}) \qquad \mathbf{u} \in \Delta = \{\mathbf{u}|\mathbf{u} \geq 0, ||\mathbf{u}||_2^2 = 1\}, \tag{8}$$

where $\hat{h}_d$ is the kernel predictor generated by $d^{th}$ base kernel $k_d$. In order to maximize model stability, the ensemble weights $\mathbf{u}$ are estimated to minimize the overall cross-validation error of $\hat{h}$. We term this method the *Cross-Validated Kernel Ensemble* (CVEK). Our proposed method belongs to the well-studied family of algorithms known as *ensembles of kernel predictors* (EKP) [7, 8, 3, 4], but with specialized focus in maximizing the algorithm's cross-validation stability. Furthermore, in addition to producing ensemble estimates $\hat{h}$, CVEK will also produce the ensemble estimate of the kernel matrix $\hat{\mathbf{K}}_0$ that is required by Algorithm 1. The exact algorithm proceeds in three stages as follows:

**Stage 1**: For each basis kernel in the library $\{k_d\}_{d=1}^D$, we first estimate $\hat{\mathbf{h}}_d = \mathbf{K}_d(\mathbf{K}_d + \hat{\lambda}_d \mathbf{I})^{-1}\mathbf{y}$, the prediction based on $d^{th}$ kernel, where the tunning parameter $\hat{\lambda}_d$ is selected by minimizing the leave-one-out cross validation (LOOCV) error [6]:

$$\texttt{LOOCV}(\lambda|k_d) = (\mathbf{I} - diag(\mathbf{A}_{d,\lambda}))^{-1}(\mathbf{y} - \hat{\mathbf{h}}_{d,\lambda}) \quad \text{where} \quad \mathbf{A}_{d,\lambda} = \mathbf{K}_d(\mathbf{K}_d + \lambda\mathbf{I})^{-1}. \tag{9}$$

We denote estimate the final LOOCV error for $d^{th}$ kernel $\hat{\epsilon}_d = \texttt{LOOCV}\left(\hat{\lambda}_d|k_d\right)$.

**Stage 2**: Using the estimated LOOCV errors $\{\hat{\epsilon}_d\}_{d=1}^D$, estimate the ensemble weights $\mathbf{u} = \{u_d\}_{d=1}^D$ such that it minimizes the overall LOOCV error:

$$\hat{\mathbf{u}} = \underset{\mathbf{u} \in \Delta}{argmin} \; ||\sum_{d=1}^D u_d \hat{\epsilon}_d||^2 \qquad \text{where} \quad \Delta = \{\mathbf{u}|\mathbf{u} \geq 0, ||\mathbf{u}||_2^2 = 1\},$$

and produce the final ensemble prediction:

$$\hat{\mathbf{h}} = \sum_{d=1}^D \hat{u}_d \mathbf{h}_d = \sum_{d=1}^D \hat{u}_d \mathbf{A}_{d,\hat{\lambda}_d}\mathbf{y} = \hat{\mathbf{A}}\mathbf{y},$$

where $\hat{\mathbf{A}} = \sum_{d=1}^D \hat{u}_d \mathbf{A}_{d,\hat{\lambda}_d}$ is the ensemble hat matrix.

**Stage 3**: Using the ensemble hat matrix $\hat{\mathbf{A}}$, estimate the ensemble kernel matrix $\hat{\mathbf{K}}$ by solving:

$$\hat{\mathbf{K}}(\hat{\mathbf{K}} + \lambda\mathbf{I})^{-1} = \hat{\mathbf{A}}.$$

Specifically, if we denote $\mathbf{U}_A$ and $\{\delta_{A,k}\}_{k=1}^n$ the eigenvector and eigenvalues of $\hat{\mathbf{A}}$, then $\hat{\mathbf{K}}$ adopts the form (see Appendix for derivation):

$$\hat{\mathbf{K}} = \mathbf{U}_A diag\left(\frac{\delta_{A,k}}{1 - \delta_{A,k}}\right)\mathbf{U}_A^T$$

## 5 Application: Testing for Nonlinear Interaction

Recall in Section 3, we assume that we are given a $k_\delta$ that generates exactly $\mathcal{H}_0$. However, depending on the exact hypothesis of interest, identifying such $k_0$ is not always straightforward. In this section,

we revisit the example about interaction testing discussed in challenge 1 at the Introduction section, and consider how to build a $k_0$ for below hypothesis of interest

$$H_0 : h(\mathbf{x}) = h_1(\mathbf{x}_1) + h_2(\mathbf{x}_2)$$
$$H_a : h(\mathbf{x}) = h_1(\mathbf{x}_1) + h_2(\mathbf{x}_2) + h_{12}(\mathbf{x}_1, \mathbf{x}_2)$$

where $h_{12}$ is the "pure interaction" function that is orthogonal to main effect functions $h_1$ and $h_2$. Recall as discussed previously, this hypothesis is difficult to formulate with Gaussian process models, since the kernel functions $k(\mathbf{x}, \mathbf{x}')$ in general do not explicitly separate the main and the interaction effect. Therefore rather than directly define $k_0$, we need to first construct $\mathcal{H}_0$ and $\mathcal{H}_a$ that corresponds to the null and alternative hypothesis, and then identify the garrote kernel function $k_\delta$ such it generates exactly $\mathcal{H}_0$ when $\delta = 0$ and $\mathcal{H}_a$ when $\delta > 0$.

We build $\mathcal{H}_0$ using the tensor-product construction of RKHS on the product domain $(\mathbf{x}_{1,i}, \mathbf{x}_{2,i}) \in \mathbb{R}^{p_1} \times \mathbb{R}^{p_2}$ [9], due to this approach's unique ability in explicitly characterizing the space of "pure interaction" functions. Let $\mathbf{1} = \{f | f \propto 1\}$ be the RKHS of constant functions, and $\mathcal{H}_1, \mathcal{H}_2$ be the RKHS of centered functions for $\mathbf{x}_1 \mathbf{x}_2$, respectively. We can then define the full space as $\mathcal{H} = \otimes_{m=1}^{2}(\mathbf{1} \oplus \mathcal{H}_m)$. $\mathcal{H}$ describes the space of functions that depends jointly on $\{\mathbf{x}_1, \mathbf{x}_2\}$, and adopts below orthogonal decomposition:

$$\mathcal{H} = (\mathbf{1} \oplus \mathcal{H}_1) \otimes (\mathbf{1} \oplus \mathcal{H}_2)$$
$$= \mathbf{1} \oplus \left\{ \mathcal{H}_1 \oplus \mathcal{H}_2 \right\} \oplus \left\{ \mathcal{H}_1 \otimes \mathcal{H}_2 \right\} = \mathbf{1} \oplus \mathcal{H}_{12}^{\perp} \oplus \mathcal{H}_{12}$$

where we have denoted $\mathcal{H}_{12}^{\perp} = \mathcal{H}_1 \oplus \mathcal{H}_2$ and $\mathcal{H}_{12} = \mathcal{H}_1 \otimes \mathcal{H}_2$, respectively. We see that $\mathcal{H}_{12}$ is indeed the space of "pure interaction" functions , since $\mathcal{H}_{12}$ contains functions on the product domain $\mathbb{R}^{p_1} \times \mathbb{R}^{p_2}$, but is orthogonal to the space of additive main effect functions $\mathcal{H}_{12}^{\perp}$. To summarize, we have identified two function spaces $\mathcal{H}_0$ and $\mathcal{H}_a$ that has the desired interpretation:

$$\mathcal{H}_0 = \mathcal{H}_{12}^{\perp} \qquad \mathcal{H}_a = \mathcal{H}_{12}^{\perp} \oplus \mathcal{H}_{12}$$

We are now ready to identify the garrote kernel $k_\delta(\mathbf{x}, \mathbf{x}')$. To this end, we notice that both $\mathcal{H}_0$ and $\mathcal{H}_{12}$ are composite spaces built from basis RKHSs using direct sum and tensor product. If denote $k_m(\mathbf{x}_m, \mathbf{x}'_m)$ the reproducing kernel associated with $\mathcal{H}_m$, we can construct kernel functions for composite spaces $\mathcal{H}_0$ and $\mathcal{H}_{12}$ as [1]:

$$k_0(\mathbf{x}, \mathbf{x}') = k_1(\mathbf{x}_1, \mathbf{x}'_1) + k_2(\mathbf{x}_2, \mathbf{x}'_2)$$
$$k_{12}(\mathbf{x}, \mathbf{x}') = k_1(\mathbf{x}_1, \mathbf{x}'_1) \, k_2(\mathbf{x}_2, \mathbf{x}'_2)$$

and consequently, the garrote kernel function for $\mathcal{H}_a$:

$$k_\delta(\mathbf{x}, \mathbf{x}') = k_0(\mathbf{x}, \mathbf{x}') + \delta * k_{12}(\mathbf{x}, \mathbf{x}'). \tag{10}$$

Finally, using the chosen form of the garrote kernel function, the $(i, j)^{th}$ element of the null derivative kernel matrix $\mathbf{K}_0$ is $\frac{\partial}{\partial \delta} k_\delta(\mathbf{x}, \mathbf{x}') = k_{12}(\mathbf{x}, \mathbf{x}')$, i.e. the null derivative kernel matrix $\partial \mathbf{K}_0$ is simply the kernel matrix $\mathbf{K}_{12}$ that corresponds to the interaction space. Therefore the score test statistic $\hat{T}_0$ in (6) simplifies to:

$$\hat{T}_0 = \hat{\tau} * (\mathbf{y} - \mathbf{X}\hat{\boldsymbol{\beta}})^T \mathbf{V}_0^{-1} \mathbf{K}_{12} \mathbf{V}_0^{-1} (\mathbf{y} - \mathbf{X}\hat{\boldsymbol{\beta}}) \tag{11}$$

where $\mathbf{V}_0 = \hat{\sigma}^2 \mathbf{I} + \hat{\tau} \mathbf{K}_0$.

## 6 Simulation Experiment

We evaluated the finite-sample performance of the proposed interaction test in a simulation study that is analogous to a real nutrition-environment interaction study. We generate two groups of input features $(\mathbf{x}_{i,1}, \mathbf{x}_{i,2}) \in \mathbb{R}^{p_1} \times \mathbb{R}^{p_2}$ independently from standard Gaussian distribution, representing normalized data representing subject's level of exposure to $p_1$ environmental pollutants and the levels of a subject's intake of $p_2$ nutrients during the study. Throughout the simulation scenarios, we keep $n = 100$, and $p_1 = p_2 = 5$. We generate the outcome $y_i$ as:

$$y_i = h_1(\mathbf{x}_{i,1}) + h_2(\mathbf{x}_{i,2}) + \delta * h_{12}(\mathbf{x}_{i,1}, \mathbf{x}_{i,2}) + \epsilon_i \tag{12}$$

where $h_1, h_2, h_{12}$ are sampled from RKHSs $\mathcal{H}_1, \mathcal{H}_2$ and $\mathcal{H}_1 \otimes \mathcal{H}_2$, generated using a ground-truth kernel $k_{\texttt{true}}$. We standardize all sampled functions to have unit norm, so that $\delta$ represents the strength of interaction relative to the main effect.

For each simulation scenario, we first generated data using $\delta$ and $k_{\texttt{true}}$ as above, then selected a $k_{\texttt{model}}$ to estimate the null model and obtain p-value using Algorithm 1. We repeated each scenario 1000 times, and evaluate the test performance using the empirical probability $\hat{P}(p \leq 0.05)$. Under null hypothesis $H_0 : \delta = 0$, $\hat{P}(p \leq 0.05)$ estimates the test's Type I error, and should be smaller or equal to the significance level 0.05. Under alternative hypothesis $H_a : \delta > 0$, $\hat{P}(p \leq 0.05)$ estimates the test's power, and should ideally approach 1 quickly as the strength of interaction $\delta$ increases.

In this study, we varied $k_{\texttt{true}}$ to produce data-generating functions $h_\delta(\mathbf{x}_{i,1}, \mathbf{x}_{i,2})$ with different smoothness and complexity properties, and varied $k_{\texttt{model}}$ to reflect different common modeling strategies for the null model in addition to using CVEK. We then evaluated how these two aspects impact the hypothesis test's Type I error and power.

**Data-generating Functions**

We sampled the data-generate function by using $k_{\texttt{true}}$ from Matérn kernel family [16]:

$$k(\mathbf{r}|\nu, \sigma) = \frac{2^{1-\nu}}{\Gamma(\nu)}\left(\sqrt{2\nu}\sigma||\mathbf{r}||\right)^\nu K_\nu\left(\sqrt{2\nu}\sigma||\mathbf{r}||\right), \qquad \text{where} \qquad \mathbf{r} = \mathbf{x} - \mathbf{x}',$$

with two non-negative hyperparameters $(\nu, \sigma)$. For a function $h$ sampled using Matérn kernel, $\nu$ determines the function's **smoothness**, since $h$ is $k$-times mean square differentiable if and only if $\nu > k$. In the case of $\nu \to \infty$, Matérn kernel reduces to the Gaussian RBF kernel which is infinitely differentiable. $\sigma$ determines the function's **complexity**, this is because in Bochner's spectral decomposition[16]

$$k(\mathbf{r}|\nu, \sigma) = \int e^{2\pi i \mathbf{s}^T \mathbf{r}} dS(\mathbf{s}|\nu, \sigma), \tag{13}$$

$\sigma$ determines how much weight the spectral density $S(\mathbf{s})$ puts on the slow-varying, low-frequency basis functions. In this work, we vary $\nu \in \{\frac{3}{2}, \frac{5}{2}, \infty\}$ to generate once-, twice-, and infinitely-differentiable functions, and vary $\sigma \in \{0.5, 1, 1.5\}$ to generate functions with varying degree of complexity.

**Modeling Strategies**

**Polynomial Kernels** is a family of simple parametric kernels that is equivalent to the polynomial ridge regression model favored by statisticians due to its model interpretability. In this work, we use the **linear** kernel $k_{\texttt{linear}}(\mathbf{x}, \mathbf{x}'|p) = \mathbf{x}^T\mathbf{x}'$ and **quadratic** kernel $k_{\texttt{quad}}(\mathbf{x}, \mathbf{x}'|p) = (1 + \mathbf{x}^T\mathbf{x}')^2$ which are common choices from this family.

**Gaussian RBF Kernels** $k_{\texttt{RBF}}(\mathbf{x}, \mathbf{x}'|\sigma) = exp(-\sigma||\mathbf{x} - \mathbf{x}'||^2)$ is a general-purpose kernel family that generates nonlinear, but infinitely differentiable (therefore very smooth) functions. Under this kernel, we consider two hyperparameter selection strategies common in machine learning applications: **RBF-Median** where we set $\sigma$ to the sample median of $\{||\mathbf{x}_i - \mathbf{x}_j||\}_{i \neq j}$, and **RBF-MLE** who estimates $\sigma$ by maximizing the model likelihood.

**Matérn and Neural Network Kernels** are two flexible kernels favored by machine learning practitioners for their expressiveness. Matérn kernels generates functions that are more flexible compared to that of Gaussian RBF due to the relaxed smoothness constraint [17]. In order to investigate the consequences of added flexibility relative to the true model, we use **Matern 1/2**, **Matern 3/2** and **Matern 5/2**, corresponding to Matérn kernels with $\nu = 1/2, 3/2,$ and $5/2$. Neural network kernels [16] $k_{\texttt{nn}}(\mathbf{x}, \mathbf{x}'|\sigma) = \frac{2}{\pi} * sin^{-1}\left(\frac{2\sigma \tilde{\mathbf{x}}^T \tilde{\mathbf{x}}'}{\sqrt{(1+2\sigma\tilde{\mathbf{x}}^T\tilde{\mathbf{x}})(1+2\sigma\tilde{\mathbf{x}}'^T\tilde{\mathbf{x}}')}}\right)$, on the other hand, represent a 1-layer Bayesian neural network with infinite hidden unit and probit link function, with $\sigma$ being the prior variance on hidden weights. Therefore $k_{\texttt{nn}}$ is flexible in the sense that it is an universal approximator for arbitrary continuous functions on the compact domain [11]. In this work, we denote **NN 0.1**, **NN 1** and **NN 10** to represent Bayesian networks with different prior constraints $\sigma \in \{0.1, 1, 10\}$.

**Result**

The simulation results are presented graphically in Figure 1 and documented in detail in the Appendix. We first observe that for reasonably specified $k_{\texttt{model}}$, the proposed hypothesis test always has the

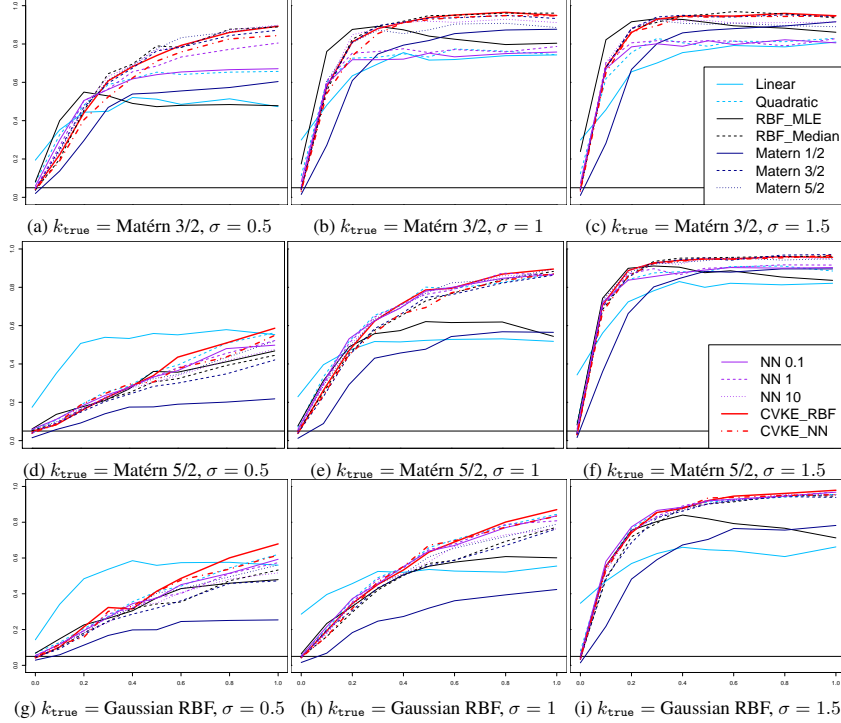

Figure 1: Estimated $\hat{P}(p < 0.05)$ (y-axis) as a function of Interaction Strength $\delta \in [0, 1]$ (x-axis).
**Skype Blue**: Linear (Solid) and Quadratic (Dashed) Kernels, **Black**: RBF-Median (Solid) and RBF-MLE (Dashed), **Dark Blue**: Matérn Kernels with $\nu = \frac{1}{2}, \frac{3}{2}, \frac{5}{2}$, **Purple**: Neural Network Kernels with $\sigma = 0.1, 1, 10$, **Red**: CVEK based on RBF (Solid) and Neural Networks (Dashed).
Horizontal line marks the test's significance level (0.05). When $\delta = 0$, $\hat{P}$ should be below this line.

correct Type I error and reasonable power. We also observe that the complexity of the data-generating function $h_\delta$ (12) plays a role in test performance, in the sense that the power of the hypothesis tests increases as the Matérn $k_{\text{true}}$'s complexity parameter $\sigma$ becomes larger, which corresponds to functions that put more weight on the complex, fast-varying eigenfunctions in (13).

We observed clear differences in test performance from different estimation strategies. In general, polynomial models (**linear** and **quadratic**) are too restrictive and appear to underfit the data under both the null and the alternative, producing inflated Type I error and diminished power. On the other hand, lower-order Matérn kernels (**Matérn 1/2** and **Matérn 3/2**, dark blue lines) appear to be too flexible. Whenever data are generated from smoother $k_{\text{true}}$, **Matérn 1/2** and **3/2** overfit the data and produce deflated Type I error and severely diminished power, even if the hyperparameter $\sigma$ are fixed at true value. Therefore unless there's strong evidence that $h$ exhibits behavior consistent with that described by these kernels, we recommend avoid the use of either polynomial or lower-order Matérn kernels for hypothesis testing. Comparatively, Gaussian RBF works well for a wider range of $k_{\text{true}}$'s, but only if the hyperparameter $\sigma$ is selected carefully. Specifically, **RBF-Median** (black dashed line) works generally well, despite being slightly conservative (i.e. lower power) when the data-generation function is smooth and of low complexity. **RBF-MLE** (black solid line), on the other hand, tends to underfit the data under the null and exhibits inflated Type I error, possibly because of the fact that $\sigma$ is not strongly identified when the sample size is too small [18]. The situation becomes more severe as $h_\delta$ becomes rougher and more complex, in the moderately extreme case of non-differentiable $h$ with $\sigma = 1.5$, the Type I error is inflated to as high as 0.238. Neural Network kernels also perform well for a wide range of $k_{\text{true}}$, and its Type I error is more robust to the specification of hyperparameters.

Finally, the two ensemble estimators **CVEK-RBF** (based on $k_{\text{RBF}}$'s with $log(\sigma) \in \{-2, -1, 0, 1, 2\}$) and **CVEK-NN** (based on $k_{\text{NN}}$'s with $\sigma \in \{0.1, 1, 10, 50\}$) perform as well or better than the non-ensemble approaches for all $k_{\text{true}}$'s, despite being slightly conservative under the null. As compared to **CVEK-NN**, **CVEK-RBF** appears to be slightly more powerful.

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
