[Supplementary Material]

# Appendix for
# Robust Hypothesis Test for Functional Effect
# with Gaussian Processes

**Jeremiah Zhe Liu, Brent Coull**
Department of Biostatistics
Harvard University
Cambridge, MA 02138
{zhl112@mail, bcoull@hsph}.harvard.edu

## Contents

# 1 Background: GP-LMM connection and REML

## 1.1 GP-LMM connection

Recall a Linear Mixed Model is formulated as:

$$\mathbf{y} = \boldsymbol{\mu} + \mathbf{h} + \boldsymbol{\epsilon} \qquad \text{where} \qquad \mathbf{h} \sim N(\mathbf{0}, \tau\mathbf{K}) \qquad \boldsymbol{\epsilon} \sim N(\mathbf{0}, \sigma^2\mathbf{I}) \tag{1}$$

Estimating $(\boldsymbol{\mu}, \mathbf{h})$ by maximizing its joint likelihood:

$$p(\mathbf{y}, \mathbf{h}|\boldsymbol{\mu}, \sigma^2, \tau) \propto p(\mathbf{y}|\mathbf{h}, \boldsymbol{\mu}, \sigma^2)p(\mathbf{h}|\tau)$$
$$= exp(-\frac{1}{2\sigma^2}||\mathbf{y} - \boldsymbol{\mu} - \mathbf{h}||^2)\, exp(-\frac{1}{2\tau}\mathbf{h}^T\mathbf{K}^{-1}\mathbf{h})$$
$$= exp\Big(-\frac{1}{2\sigma^2}||\mathbf{y} - \boldsymbol{\mu} - \mathbf{h}||^2 - \frac{1}{2\tau}\mathbf{h}^T\mathbf{K}^{-1}\mathbf{h}\Big)$$

or equivalently, by minimizing the negative log likelihood:

$$-log\, p(\mathbf{y}, \mathbf{h}|\boldsymbol{\mu}, \sigma^2, \tau) = \frac{1}{2\sigma^2}||\mathbf{y} - \boldsymbol{\mu} - \mathbf{h}||^2 + \frac{1}{2\tau}\mathbf{h}^T\mathbf{K}^{-1}\mathbf{h}$$
$$\propto ||\mathbf{y} - \boldsymbol{\mu} - \mathbf{h}||^2 + \frac{\sigma}{\tau}\mathbf{h}^T\mathbf{K}^{-1}\mathbf{h} \tag{2}$$

Fixing $(\sigma^2, \tau)$, minimizer of above objective function is :

$$\widehat{\boldsymbol{\mu}} = (\mathbf{1}^T\mathbf{V}^{-1}\mathbf{1})^{-1}\mathbf{1}^T\mathbf{V}^{-1}\mathbf{y} \tag{3}$$
$$\widehat{\mathbf{h}} = \mathbf{K}\mathbf{V}^{-1}(\mathbf{y} - \boldsymbol{\mu}^*) \tag{4}$$

where $\mathbf{V} = \mathbf{I} + \frac{\tau}{\sigma^2}\mathbf{K}$. Notice that if we set $\lambda = \frac{\tau}{\sigma^2}$, the expression of $\widehat{\mathbf{h}}$ becomes:

$$\widehat{\mathbf{h}} = \mathbf{K}(\mathbf{K} + \lambda\mathbf{I})^{-1}(\mathbf{y} - \mathbf{u}^*),$$

which correspond exactly to the posterior predictive mean of $\mathbf{h}$ in Gaussian Process model.

## 1.2 REstricted Maximum Likelihood (REML) for variance parameters

*Motivation: MLE variance estimators are biased in small sample*

Given the $(\widehat{\boldsymbol{\mu}}, \widehat{\mathbf{h}})$ estimates in (3)-(4), it is tempting to estimate the variance parameters $(\tau, \sigma^2)$ by directly minimizing (2) while naively fixing $(\mu, \mathbf{h})$ to the estimated values $(\widehat{\mu}, \widehat{\mathbf{h}})$. However, this practice fail to acknowledge the fact that $\widehat{\boldsymbol{\mu}}$ is estimated from the data, and consequently produces biased estimates for $(\tau, \sigma^2)$. To see why this is the case, consider the simplified case where $\mathbf{K} = \mathbf{I}$ and $\sigma^2$ is known and fixed, such that (1) corresponds to a linear regression model

$$\mathbf{y} = \boldsymbol{\mu} + \boldsymbol{\epsilon} \quad \text{where} \quad \boldsymbol{\epsilon} \sim N(\mathbf{0}, (\sigma^2 + \tau)\mathbf{I}),$$

and $\tau$ is estimated by minimizing the corresponding negative log likelihood:

$$-log\, p(\mathbf{y}|\widehat{\boldsymbol{\mu}}, \sigma^2, \tau) = n * log(\sigma^2 + \tau) + \frac{(\mathbf{y} - \widehat{\boldsymbol{\mu}})^T(\mathbf{y} - \widehat{\boldsymbol{\mu}})}{(\sigma^2 + \tau)},$$

with solution

$$\widehat{\tau} = \frac{(\mathbf{y} - \widehat{\boldsymbol{\mu}})^T(\mathbf{y} - \widehat{\boldsymbol{\mu}})}{n} - \sigma^2,$$

The expectation of $\widehat{\tau}$ is:

$$E(\widehat{\tau}) = \frac{1}{n}E\Big((\mathbf{y} - \widehat{\boldsymbol{\mu}})^T(\mathbf{y} - \widehat{\boldsymbol{\mu}})\Big) - \sigma^2 = \frac{n-1}{n}(\tau + \sigma^2) - \sigma^2 = \tau - \frac{1}{n}(\tau + \sigma^2)$$

where we have used the fact that $(\mathbf{y} - \widehat{\boldsymbol{\mu}})^T(\mathbf{y} - \widehat{\boldsymbol{\mu}}) \sim (\sigma^2 + \tau) * \chi^2_{n-1}$. Here the degrees of freedom of $\chi^2$ distribution is $n - 1$ instead of $n$ as a result of the fact that $\widehat{\boldsymbol{\mu}}$ is estimated from the data.

Consequently, when the sample size $n$ is small and variance $\sigma^2$ is large, we see that $\widehat{\tau}$ is a biased estimate of $\tau$. Since also $\widehat{\lambda} = \frac{\widehat{\tau}}{\sigma^2}$ (recall $\sigma^2$ is known), we underestimate $\lambda$ in finite sample, leading to insufficient regularization in small sample sizes.

*Estimating $(\tau, \sigma^2)$ unbiasedly using REML*

In order to tackle the above complications, Harville [4] introduced REML (REstricted Maximum Likelihood or REsidual Maximum Likelihood) to eliminate the presence of $(\boldsymbol{\mu}, \mathbf{h})$ from the likelihood, such that we no longer need to plug in the estimated parameters $(\widehat{\boldsymbol{\mu}}, \widehat{\mathbf{h}})$ in place of the true parameters when estimating $(\tau, \sigma^2)$. Specifically, recall that under the LMM (1), $\mathbf{y} \sim N(\boldsymbol{\mu} = \mu * \mathbf{1}, \mathbf{V})$ where $\mathbf{V} = \sigma^2 \mathbf{I} + \tau \mathbf{K}$, and there always exists a matrix $\mathbf{A}$ satisfying $\mathbf{A}\mathbf{1} = \mathbf{0}$ such that $\mathbf{A}\mathbf{y} \sim N(\mathbf{A}\boldsymbol{\mu} = \mathbf{0}, \mathbf{A}\mathbf{V}\mathbf{A}^T)$. Notice that the distribution of $\mathbf{A}\mathbf{y} \sim N(\mathbf{0}, \mathbf{A}(\sigma^2\mathbf{I} + \tau\mathbf{K})\mathbf{A}^T)$ no longer contains the nuisance parameter $\mu$. Therefore if we estimate $(\sigma^2, \tau)$ by maximizing the *REstricted likelihood* $p(\mathbf{A}\mathbf{y})$:

$$-log\, p(\mathbf{A}\mathbf{y}|\sigma^2, \tau) = |\mathbf{A}\mathbf{V}\mathbf{A}^T| + \mathbf{y}^T\mathbf{A}(\mathbf{A}\mathbf{V}\mathbf{A}^T)^{-1}\mathbf{A}\mathbf{y} \tag{5}$$

instead of the naive likelihood $p(\mathbf{y})$, we no longer need to plug in the estimated parameters $\widehat{\boldsymbol{\mu}}$ to the loss function. Furthermore, [4] has shown that when $\mathbf{A}^T\mathbf{A} = \mathbf{I}$, (5) can be expressed as:

$$-logp(\mathbf{A}\mathbf{y}|\sigma^2, \tau) = log|\mathbf{V}| + log|\mathbf{1}^T\mathbf{V}^{-1}\mathbf{1}| + (\mathbf{y} - \boldsymbol{\mu})^T\mathbf{V}^{-1}(\mathbf{y} - \boldsymbol{\mu}),$$

giving rise to the expression of the REML likelihood for Linear Mixed Models.

## 2 Derivation for Null Distribution

In this section, we derive the null distribution for the test statistic:

$$\widehat{T}_0 = \widehat{\tau} * (\mathbf{y} - \mathbf{X}\widehat{\boldsymbol{\beta}})^T\mathbf{V}_0^{-1}\, \partial\mathbf{K}_0\, \mathbf{V}_0^{-1}(\mathbf{y} - \mathbf{X}\widehat{\boldsymbol{\beta}}), \tag{6}$$

where $\mathbf{V}_\delta = \tau\mathbf{K}_\delta + \sigma^2\mathbf{I}$.

We first derive the first two moments of $\widehat{T}_0$ under $H_0 : \delta = 0$. First derive the expectation:

$$
\begin{aligned}
E(T_0) &= E\Big(\widehat{\tau} * (\mathbf{y} - \mathbf{X}\widehat{\boldsymbol{\beta}})^T\mathbf{V}_0^{-1}\, \partial\mathbf{K}_0\, \mathbf{V}_0^{-1}(\mathbf{y} - \mathbf{X}\widehat{\boldsymbol{\beta}})\Big) \\
&= \widehat{\tau} * E\Big(tr\big((\mathbf{y} - \mathbf{X}\widehat{\boldsymbol{\beta}})^T\mathbf{V}_0^{-1}\, \partial\mathbf{K}_0\, \mathbf{V}_0^{-1}(\mathbf{y} - \mathbf{X}\widehat{\boldsymbol{\beta}})\big)\Big) \\
&= \widehat{\tau} * E\Big(tr\big(\mathbf{V}_0^{-1}\, \partial\mathbf{K}_0\, \mathbf{V}_0^{-1}(\mathbf{y} - \mathbf{X}\widehat{\boldsymbol{\beta}})(\mathbf{y} - \mathbf{X}\widehat{\boldsymbol{\beta}})^T\big)\Big) \\
&= \widehat{\tau} * tr\Big(\mathbf{V}_0^{-1}\, \partial\mathbf{K}_0\, \mathbf{V}_0^{-1}E\big((\mathbf{y} - \mathbf{X}\widehat{\boldsymbol{\beta}})(\mathbf{y} - \mathbf{X}\widehat{\boldsymbol{\beta}})^T\big)\Big) \\
&= \widehat{\tau} * tr\Big(\mathbf{V}_0^{-1}\, \partial\mathbf{K}_0\, \mathbf{V}_0^{-1}\mathbf{V}_0\Big) \\
&= \widehat{\tau} * tr\Big(\mathbf{V}_0^{-1}\, \partial\mathbf{K}_0\Big).
\end{aligned}
$$

By maximum likelihood theory, the asymptotic variance of the score statistic for $\delta$ is the corresponding submatrix of the inverse Fisher information matrix (Hessian of the model likelihood). If denote $\boldsymbol{\omega} = [\mu, \sigma, \tau]$ the model's offset and variance component parameters, and $\boldsymbol{\theta} = [\boldsymbol{\omega}, \delta]$ the vector of all model parameters, then the information matrix takes the following form:

$$\mathbf{I} = \begin{bmatrix} \mathbf{I}_{\boldsymbol{\omega\omega}} & \mathbf{I}_{\boldsymbol{\omega}\delta} \\ \mathbf{I}_{\delta\boldsymbol{\omega}} & \mathbf{I}_{\delta\delta} \end{bmatrix}.$$

Under REML, the $(i, j)^{th}$ of $\mathbf{I}$ can be expressed as [8]

$$\mathbf{I}_{\theta_i\theta_j} = \frac{\partial}{\partial\theta_i\theta_j}L_{\texttt{MLE}}(\boldsymbol{\theta}) = tr\Big(\mathbf{P}_0\, (\frac{\partial}{\partial\theta_i}\mathbf{V}_0)\, \mathbf{P}_0\, (\frac{\partial}{\partial\theta_j}\mathbf{V}_0)\Big)$$

where $\mathbf{P}_0 = \mathbf{V}^{-1} - \mathbf{V}^{-1}\mathbf{X}^T(\mathbf{X}\mathbf{V}^{-1}\mathbf{X}^T)^{-1}\mathbf{X}\mathbf{V}^{-1}$ is the "scaled projection matrix" under REML, such that $\mathbf{P}_0\mathbf{y} = \frac{1}{\sigma^2}(\mathbf{y} - \mathbf{X}\widehat{\boldsymbol{\beta}} - \widehat{\mathbf{h}})$ under the correct model. Consequently, the variance of $T_0$ is the

submatrix of $\mathbf{I}^{-1}$ that corresponds to the parameter $\delta$. Using the block matrix inversion formula, we can express $Var(T_0)$ as:

$$Var(T_0) = \widehat{\mathbf{i}}_{\delta\delta} = \mathbf{I}_{\delta\delta} - \mathbf{I}_{\delta\omega}^T \mathbf{I}_{\omega\omega}^{-1} \mathbf{I}_{\delta\omega}$$

where $\widehat{\mathbf{i}}_{\delta\delta}$ is commonly referred to as "efficient information" [7].

Consequently, the estimation equations for the method of moment estimator of $(\kappa, \nu)$ in the Satterthwaite method can be expressed as:

$$\kappa * \nu = E(T) = \hat{\tau} * tr\left(\mathbf{V}_0^{-1} \, \partial\mathbf{K}_0\right), \qquad 2 * \kappa^2 * \nu = Var(T) = \widehat{\mathbf{i}}_{\delta\delta}$$

with solution:

$$\widehat{\kappa} = \widehat{\mathbf{i}}_{\delta\delta} / \left[\hat{\tau} * tr\left(\mathbf{V}_0^{-1} \partial\mathbf{K}_0\right)\right]$$

$$\widehat{\nu} = \left[\hat{\tau} * tr\left(\mathbf{V}_0^{-1} \partial\mathbf{K}_0\right)\right]^2 / (2 * \widehat{\mathbf{I}}_{\delta\delta})$$

# 3 Derivation for Ensemble Kernel Matrix

Given the ensemble hat matrix $\widehat{\mathbf{A}}$ in Section 4, we consider how to identify the ensemble kernel matrix $\widehat{\mathbf{K}}$ by solving:

$$\widehat{\mathbf{K}}(\widehat{\mathbf{K}} + \lambda\mathbf{I})^{-1} = \widehat{\mathbf{A}}.$$

Specifically, if denote $(\mathbf{U}_A, \mathbf{U}_K)$ and $(\{\delta_{A,k}\}_{k=1}^n, \{\delta_{K,k}\}_{k=1}^n)$ the eigenvector and eigenvalues of $\widehat{\mathbf{A}}$ and $\widehat{\mathbf{K}}$, respectively, then the above system reduces to:

$$\mathbf{U}_A diag\left(\delta_{A,k}\right)\mathbf{U}_A^T = \mathbf{U}_K diag\left(\frac{\delta_{K,k}}{\delta_{K,k} + \lambda}\right)\mathbf{U}_K^T$$

and adopts closed form solution $\mathbf{U}_K = \mathbf{U}_A$ and $\delta_{K,k} = \lambda\frac{\delta_{A,k}}{1 - \delta_{A,k}}$. Therefore the ensemble kernel matrix $\widehat{\mathbf{K}}$ is estimated as:

$$\widehat{\mathbf{K}} = \lambda * \mathbf{U}_A diag\left(\frac{\delta_{A,k}}{1 - \delta_{A,k}}\right)\mathbf{U}_A^T.$$

Notice that we have left the "ensemble tunning parameter" $\lambda$ unspecified. In practice, $\lambda$ serves only as a constant scaling factor for the kernel matrix $\mathbf{K}$, whose exact value does not impact either the prediction or the p-value calculation, since both procedures are scale invariant with respect to the kernel matrix. We therefore trivially set $\lambda = 1$, leading to the final estimate for ensemble kernel matrix:

$$\widehat{\mathbf{K}} = \mathbf{U}_A diag\left(\frac{\delta_{A,k}}{1 - \delta_{A,k}}\right)\mathbf{U}_A^T.$$

# 4 Simulation Results

In this section we document the value of estimated $\widehat{P}(p < 0.05)$ from the simulation presented in Section 6 (Simulation Experiment) of the paper. Recall that the simulation data is generated from below mechanism:

$$y_i = h_1(\mathbf{x}_{i,1}) + h_2(\mathbf{x}_{i,2}) + \delta * h_{12}(\mathbf{x}_{i,1}, \mathbf{x}_{i,2}) + \epsilon_i \tag{7}$$

where $h_i$'s are functions with unit norm sampled from the reproducing kernel Hilbert spaces (RKHSs) generated by $k_{\text{true}}$, and the data is then fitted using Gaussian process with $k_{\text{model}}$.

Each table documents the $\widehat{P}(p < 0.05)$ resulted from fixing $k_{\text{true}}$ to a Matérn kernel with specific value of smoothness parameter $\nu$ and complexity parameter $\sigma$, and then varying the strength of the interaction $\delta \in [0, 1]$ and the model kernel $k_{\text{model}}$.

Our general observations are:

1. The value of test power increases as the value $k_{\text{true}}$'s complexity parameter $\sigma$ becomes larger. This is possibly caused by the fact that the interaction becomes easier to detect as the pure interaction function $h_{12} \in \mathcal{H}_{12}$ becomes more complex as in it varies more quickly.

2. Given the data-generation mechanism:

    (a) Polynomial kernels (Linear and Quadratic kernels) exhibits underfit, and result in inflated Type I error but also low power.

    (b) Lower-order Matérn kernels (Matern 1/2 and 3/2) tend to exhibits overfit for smoother $k_{\text{true}}$'s, and result in deflated Type I error and diminished low power. This conclusion cautions us against the approach of extending model complexity by naively relaxing model's smoothness (i.e. differentiability) constraint.

    (c) Gaussian RBF Kernels in general can perform well, but only if the hyperparameter is chosen carefully. Specifically, selecting the hyperparameter $\sigma$ by maximizing model likelihood does not perform well in small sample. On the other hand, the naive approach of selecting $\sigma$ by setting $\sigma$ to population median performs surprisingly well.

    (d) Neural Network kernels also work well in general. Their performance is also impacted by the hyperparameters. However not as sensitive as Gaussian RBF.

Table 1: $k_{\text{true}} = $ Matérn 3/2, $\sigma = 0.5$

| $k_{\text{model}}/\delta$ | 0 | 0.1 | 0.2 | 0.3 | 0.4 | 0.5 | 0.6 | 0.8 | 1 |
|---|---|---|---|---|---|---|---|---|---|
| Linear | 0.194 | 0.352 | 0.444 | 0.449 | 0.521 | 0 | 0.485 | 0.514 | 0.473 |
| Quadratic | 0.078 | 0.326 | 0.481 | 0.588 | 0.619 | 0.653 | 0.641 | 0.653 | 0.657 |
| RBF_MLE | 0.081 | 0.400 | 0.549 | 0.529 | 0.490 | 0.473 | 0.480 | 0.484 | 0.478 |
| RBF_Median | 0.038 | 0.199 | 0.453 | 0.645 | 0.695 | 0.790 | 0.782 | 0.877 | 0.889 |
| Matern 1/2 | 0.020 | 0.137 | 0.296 | 0.469 | 0.539 | 0.545 | 0.555 | 0.573 | 0.604 |
| Matern 3/2 | 0.047 | 0.251 | 0.471 | 0.596 | 0.674 | 0.764 | 0.783 | 0.844 | 0.872 |
| Matern 5/2 | 0.035 | 0.243 | 0.458 | 0.612 | 0.688 | 0.775 | 0.833 | 0.863 | 0.896 |
| NN 0.1 | 0.059 | 0.299 | 0.505 | 0.563 | 0.618 | 0.640 | 0.654 | 0.666 | 0.671 |
| NN 1 | 0.047 | 0.266 | 0.504 | 0.565 | 0.655 | 0.685 | 0.732 | 0.772 | 0.805 |
| NN 10 | 0.050 | 0.234 | 0.477 | 0.631 | 0.687 | 0.769 | 0.786 | 0.874 | 0.898 |
| CVKE_RBF | 0.044 | 0.222 | 0.441 | 0.607 | 0.682 | 0.740 | 0.792 | 0.860 | 0.893 |
| CVKE_NN | 0.041 | 0.190 | 0.405 | 0.524 | 0.622 | 0.711 | 0.758 | 0.826 | 0.844 |

Table 2: $k_{\text{true}} = $ Matérn 3/2, $\sigma = 1$

| $k_{\text{model}}/\delta$ | 0 | 0.1 | 0.2 | 0.3 | 0.4 | 0.5 | 0.6 | 0.8 | 1 |
|---|---|---|---|---|---|---|---|---|---|
| Linear | 0.299 | 0.481 | 0.634 | 0.696 | 0.755 | 0.716 | 0.719 | 0.739 | 0.743 |
| Quadratic | 0.113 | 0.603 | 0.726 | 0.731 | 0.749 | 0.732 | 0.774 | 0.762 | 0.744 |
| RBF_MLE | 0.174 | 0.761 | 0.876 | 0.892 | 0.874 | 0.841 | 0.825 | 0.797 | 0.804 |
| RBF_Median | 0.045 | 0.556 | 0.825 | 0.893 | 0.919 | 0.948 | 0.950 | 0.961 | 0.961 |
| Matern 1/2 | 0.015 | 0.272 | 0.609 | 0.748 | 0.794 | 0.818 | 0.854 | 0.873 | 0.877 |
| Matern 3/2 | 0.044 | 0.574 | 0.808 | 0.896 | 0.914 | 0.935 | 0.936 | 0.949 | 0.933 |
| Matern 5/2 | 0.040 | 0.606 | 0.807 | 0.873 | 0.854 | 0.874 | 0.904 | 0.908 | 0.886 |
| NN 0.1 | 0.081 | 0.593 | 0.718 | 0.718 | 0.721 | 0.752 | 0.733 | 0.750 | 0.758 |
| NN 1 | 0.058 | 0.608 | 0.752 | 0.761 | 0.775 | 0.755 | 0.771 | 0.759 | 0.787 |
| NN 10 | 0.046 | 0.578 | 0.848 | 0.880 | 0.913 | 0.919 | 0.913 | 0.929 | 0.912 |
| CVKE_RBF | 0.041 | 0.578 | 0.811 | 0.881 | 0.912 | 0.938 | 0.951 | 0.965 | 0.949 |
| CVKE_NN | 0.032 | 0.541 | 0.738 | 0.854 | 0.911 | 0.927 | 0.939 | 0.948 | 0.947 |

Table 3: $k_{\texttt{true}} = $ Matérn 3/2, $\sigma = 1.5$

| $k_{\texttt{model}}/\delta$ | 0 | 0.1 | 0.2 | 0.3 | 0.4 | 0.5 | 0.6 | 0.8 | 1 |
|---|---|---|---|---|---|---|---|---|---|
| Linear | 0.299 | 0.457 | 0.655 | 0.701 | 0.755 | 0.772 | 0.792 | 0.785 | 0.810 |
| Quadratic | 0.123 | 0.619 | 0.756 | 0.829 | 0.806 | 0.788 | 0.815 | 0.812 | 0.830 |
| RBF_MLE | 0.239 | 0.822 | 0.916 | 0.932 | 0.928 | 0.912 | 0.895 | 0.882 | 0.861 |
| RBF_Median | 0.040 | 0.676 | 0.881 | 0.913 | 0.947 | 0.955 | 0.969 | 0.957 | 0.936 |
| Matern 1/2 | 0.010 | 0.282 | 0.667 | 0.802 | 0.858 | 0.872 | 0.880 | 0.894 | 0.916 |
| Matern 3/2 | 0.043 | 0.675 | 0.880 | 0.941 | 0.950 | 0.942 | 0.941 | 0.952 | 0.943 |
| Matern 5/2 | 0.041 | 0.678 | 0.883 | 0.923 | 0.908 | 0.903 | 0.909 | 0.896 | 0.890 |
| NN 0.1 | 0.073 | 0.671 | 0.785 | 0.801 | 0.788 | 0.817 | 0.797 | 0.822 | 0.806 |
| NN 1 | 0.046 | 0.703 | 0.806 | 0.817 | 0.811 | 0.817 | 0.815 | 0.790 | 0.828 |
| NN 10 | 0.031 | 0.702 | 0.881 | 0.933 | 0.919 | 0.922 | 0.910 | 0.911 | 0.915 |
| CVKE_RBF | 0.042 | 0.681 | 0.860 | 0.930 | 0.945 | 0.947 | 0.946 | 0.960 | 0.947 |
| CVKE_NN | 0.034 | 0.650 | 0.863 | 0.895 | 0.942 | 0.941 | 0.944 | 0.944 | 0.946 |

Table 4: $k_{\texttt{true}} = $ Matérn 5/2, $\sigma = 0.5$

| $k_{\texttt{model}}/\delta$ | 0 | 0.1 | 0.2 | 0.3 | 0.4 | 0.5 | 0.6 | 0.8 | 1 |
|---|---|---|---|---|---|---|---|---|---|
| Linear | 0.174 | 0.351 | 0.507 | 0.539 | 0.533 | 0.559 | 0.552 | 0.579 | 0.553 |
| Quadratic | 0.055 | 0.107 | 0.186 | 0.253 | 0.284 | 0.359 | 0.394 | 0.508 | 0.563 |
| RBF_MLE | 0.061 | 0.137 | 0.174 | 0.209 | 0.270 | 0.361 | 0.357 | 0.411 | 0.468 |
| RBF_Median | 0.052 | 0.091 | 0.162 | 0.214 | 0.250 | 0.306 | 0.323 | 0.392 | 0.445 |
| Matern 1/2 | 0.015 | 0.058 | 0.092 | 0.140 | 0.175 | 0.176 | 0.190 | 0.201 | 0.218 |
| Matern 3/2 | 0.041 | 0.089 | 0.148 | 0.203 | 0.242 | 0.283 | 0.300 | 0.348 | 0.421 |
| Matern 5/2 | 0.056 | 0.099 | 0.154 | 0.222 | 0.275 | 0.323 | 0.345 | 0.433 | 0.519 |
| NN 0.1 | 0.059 | 0.111 | 0.178 | 0.235 | 0.277 | 0.332 | 0.365 | 0.480 | 0.498 |
| NN 1 | 0.038 | 0.091 | 0.161 | 0.224 | 0.281 | 0.332 | 0.380 | 0.455 | 0.522 |
| NN 10 | 0.039 | 0.113 | 0.165 | 0.213 | 0.271 | 0.304 | 0.339 | 0.418 | 0.476 |
| CVKE_RBF | 0.049 | 0.083 | 0.155 | 0.221 | 0.279 | 0.339 | 0.435 | 0.509 | 0.586 |
| CVKE_NN | 0.039 | 0.085 | 0.186 | 0.245 | 0.295 | 0.306 | 0.377 | 0.436 | 0.549 |

Table 5: $k_{\texttt{true}} = $ Matérn 5/2, $\sigma = 1$

| $k_{\texttt{model}}/\delta$ | 0 | 0.1 | 0.2 | 0.3 | 0.4 | 0.5 | 0.6 | 0.8 | 1 |
|---|---|---|---|---|---|---|---|---|---|
| Linear | 0.229 | 0.396 | 0.471 | 0.517 | 0.515 | 0.523 | 0.528 | 0.531 | 0.518 |
| Quadratic | 0.071 | 0.333 | 0.517 | 0.654 | 0.703 | 0.801 | 0.793 | 0.825 | 0.869 |
| RBF_MLE | 0.077 | 0.313 | 0.489 | 0.558 | 0.574 | 0.621 | 0.616 | 0.619 | 0.544 |
| RBF_Median | 0.050 | 0.251 | 0.455 | 0.576 | 0.648 | 0.729 | 0.767 | 0.840 | 0.883 |
| Matern 1/2 | 0.012 | 0.089 | 0.292 | 0.430 | 0.457 | 0.477 | 0.543 | 0.568 | 0.565 |
| Matern 3/2 | 0.039 | 0.230 | 0.444 | 0.584 | 0.657 | 0.748 | 0.761 | 0.822 | 0.863 |
| Matern 5/2 | 0.052 | 0.287 | 0.475 | 0.636 | 0.692 | 0.770 | 0.823 | 0.842 | 0.896 |
| NN 0.1 | 0.059 | 0.303 | 0.531 | 0.626 | 0.691 | 0.780 | 0.799 | 0.847 | 0.867 |
| NN 1 | 0.052 | 0.292 | 0.508 | 0.645 | 0.708 | 0.763 | 0.785 | 0.874 | 0.865 |
| NN 10 | 0.043 | 0.299 | 0.493 | 0.624 | 0.693 | 0.785 | 0.787 | 0.860 | 0.869 |
| CVKE_RBF | 0.037 | 0.263 | 0.470 | 0.623 | 0.710 | 0.786 | 0.795 | 0.869 | 0.895 |
| CVKE_NN | 0.049 | 0.237 | 0.449 | 0.563 | 0.660 | 0.694 | 0.771 | 0.835 | 0.871 |

Table 6: $k_{\texttt{true}} = $ Matérn 5/2, $\sigma = 1.5$

| $k_{\texttt{model}}/\delta$ | 0 | 0.1 | 0.2 | 0.3 | 0.4 | 0.5 | 0.6 | 0.8 | 1 |
|---|---|---|---|---|---|---|---|---|---|
| Linear | 0.343 | 0.561 | 0.724 | 0.782 | 0.830 | 0.800 | 0.821 | 0.813 | 0.821 |
| Quadratic | 0.096 | 0.723 | 0.840 | 0.875 | 0.881 | 0.881 | 0.908 | 0.911 | 0.885 |
| RBF_MLE | 0.082 | 0.743 | 0.899 | 0.911 | 0.905 | 0.876 | 0.886 | 0.854 | 0.836 |
| RBF_Median | 0.038 | 0.684 | 0.858 | 0.935 | 0.952 | 0.954 | 0.956 | 0.961 | 0.962 |
| Matern 1/2 | 0.016 | 0.360 | 0.663 | 0.802 | 0.846 | 0.883 | 0.879 | 0.896 | 0.896 |
| Matern 3/2 | 0.034 | 0.698 | 0.853 | 0.925 | 0.944 | 0.952 | 0.941 | 0.968 | 0.971 |
| Matern 5/2 | 0.046 | 0.733 | 0.877 | 0.921 | 0.930 | 0.939 | 0.954 | 0.942 | 0.949 |
| NN 0.1 | 0.059 | 0.721 | 0.837 | 0.856 | 0.875 | 0.890 | 0.903 | 0.900 | 0.903 |
| NN 1 | 0.039 | 0.700 | 0.870 | 0.897 | 0.865 | 0.899 | 0.904 | 0.917 | 0.916 |
| NN 10 | 0.044 | 0.729 | 0.888 | 0.928 | 0.920 | 0.953 | 0.948 | 0.960 | 0.946 |
| CVKE_RBF | 0.031 | 0.708 | 0.887 | 0.928 | 0.940 | 0.947 | 0.948 | 0.960 | 0.957 |
| CVKE_NN | 0.032 | 0.671 | 0.859 | 0.925 | 0.935 | 0.949 | 0.946 | 0.954 | 0.966 |

Table 7: $k_{\texttt{true}} = $ Gaussian RBF, $\sigma = 0.5$

| $k_{\texttt{model}}/\delta$ | 0 | 0.1 | 0.2 | 0.3 | 0.4 | 0.5 | 0.6 | 0.8 | 1 |
|---|---|---|---|---|---|---|---|---|---|
| Linear | 0.143 | 0.341 | 0.484 | 0.537 | 0.585 | 0.559 | 0.574 | 0.576 | 0.560 |
| Quadratic | 0.069 | 0.128 | 0.206 | 0.273 | 0.355 | 0.412 | 0.443 | 0.545 | 0.620 |
| RBF_MLE | 0.068 | 0.148 | 0.226 | 0.263 | 0.305 | 0.377 | 0.429 | 0.459 | 0.479 |
| RBF_Median | 0.045 | 0.100 | 0.181 | 0.245 | 0.318 | 0.343 | 0.354 | 0.473 | 0.533 |
| Matern 1/2 | 0.029 | 0.059 | 0.113 | 0.167 | 0.198 | 0.199 | 0.245 | 0.251 | 0.254 |
| Matern 3/2 | 0.045 | 0.092 | 0.171 | 0.247 | 0.286 | 0.320 | 0.361 | 0.458 | 0.472 |
| Matern 5/2 | 0.046 | 0.124 | 0.181 | 0.271 | 0.319 | 0.403 | 0.427 | 0.495 | 0.561 |
| NN 0.1 | 0.054 | 0.118 | 0.194 | 0.268 | 0.345 | 0.375 | 0.451 | 0.515 | 0.593 |
| NN 1 | 0.055 | 0.118 | 0.194 | 0.287 | 0.322 | 0.379 | 0.402 | 0.513 | 0.574 |
| NN 10 | 0.042 | 0.103 | 0.184 | 0.239 | 0.335 | 0.348 | 0.407 | 0.482 | 0.517 |
| CVKE_RBF | 0.041 | 0.103 | 0.215 | 0.323 | 0.315 | 0.414 | 0.486 | 0.601 | 0.679 |
| CVKE_NN | 0.044 | 0.117 | 0.157 | 0.301 | 0.330 | 0.411 | 0.477 | 0.538 | 0.616 |

Table 8: $k_{\texttt{true}} = $ Gaussian RBF, $\sigma = 1$

| $k_{\texttt{model}}/\delta$ | 0 | 0.1 | 0.2 | 0.3 | 0.4 | 0.5 | 0.6 | 0.8 | 1 |
|---|---|---|---|---|---|---|---|---|---|
| Linear | 0.286 | 0.396 | 0.457 | 0.525 | 0.520 | 0.536 | 0.527 | 0.521 | 0.555 |
| Quadratic | 0.056 | 0.203 | 0.369 | 0.490 | 0.546 | 0.658 | 0.702 | 0.783 | 0.844 |
| RBF_MLE | 0.065 | 0.234 | 0.330 | 0.430 | 0.507 | 0.554 | 0.577 | 0.608 | 0.601 |
| RBF_Median | 0.046 | 0.161 | 0.297 | 0.421 | 0.502 | 0.570 | 0.587 | 0.693 | 0.772 |
| Matern 1/2 | 0.016 | 0.068 | 0.183 | 0.247 | 0.273 | 0.320 | 0.361 | 0.394 | 0.424 |
| Matern 3/2 | 0.042 | 0.198 | 0.307 | 0.433 | 0.504 | 0.558 | 0.588 | 0.670 | 0.764 |
| Matern 5/2 | 0.043 | 0.184 | 0.340 | 0.458 | 0.510 | 0.607 | 0.655 | 0.720 | 0.789 |
| NN 0.1 | 0.053 | 0.216 | 0.373 | 0.456 | 0.552 | 0.639 | 0.670 | 0.770 | 0.836 |
| NN 1 | 0.045 | 0.185 | 0.354 | 0.481 | 0.545 | 0.647 | 0.678 | 0.788 | 0.808 |
| NN 10 | 0.044 | 0.175 | 0.347 | 0.465 | 0.510 | 0.579 | 0.664 | 0.730 | 0.763 |
| CVKE_RBF | 0.043 | 0.193 | 0.346 | 0.452 | 0.533 | 0.633 | 0.690 | 0.801 | 0.870 |
| CVKE_NN | 0.043 | 0.162 | 0.318 | 0.467 | 0.552 | 0.671 | 0.696 | 0.772 | 0.834 |

Table 9: $k_{\mathtt{true}}$ = Gaussian RBF, $\sigma = 1.5$

| $k_{\mathtt{model}}/\delta$ | 0 | 0.1 | 0.2 | 0.3 | 0.4 | 0.5 | 0.6 | 0.8 | 1 |
|---|---|---|---|---|---|---|---|---|---|
| Linear | 0.347 | 0.471 | 0.569 | 0.625 | 0.660 | 0.646 | 0.640 | 0.608 | 0.662 |
| Quadratic | 0.080 | 0.554 | 0.767 | 0.854 | 0.883 | 0.913 | 0.922 | 0.941 | 0.956 |
| RBF_MLE | 0.052 | 0.555 | 0.755 | 0.804 | 0.840 | 0.819 | 0.792 | 0.766 | 0.712 |
| RBF_Median | 0.046 | 0.481 | 0.719 | 0.795 | 0.882 | 0.902 | 0.914 | 0.950 | 0.946 |
| Matern 1/2 | 0.014 | 0.218 | 0.482 | 0.591 | 0.673 | 0.704 | 0.765 | 0.756 | 0.782 |
| Matern 3/2 | 0.036 | 0.494 | 0.686 | 0.825 | 0.862 | 0.903 | 0.920 | 0.943 | 0.939 |
| Matern 5/2 | 0.047 | 0.543 | 0.755 | 0.829 | 0.869 | 0.903 | 0.925 | 0.946 | 0.955 |
| NN 0.1 | 0.054 | 0.581 | 0.774 | 0.866 | 0.884 | 0.919 | 0.929 | 0.946 | 0.968 |
| NN 1 | 0.037 | 0.553 | 0.750 | 0.834 | 0.891 | 0.923 | 0.926 | 0.947 | 0.969 |
| NN 10 | 0.044 | 0.523 | 0.756 | 0.827 | 0.877 | 0.901 | 0.922 | 0.956 | 0.950 |
| CVKE_RBF | 0.034 | 0.554 | 0.741 | 0.855 | 0.877 | 0.921 | 0.946 | 0.961 | 0.979 |
| CVKE_NN | 0.043 | 0.534 | 0.749 | 0.855 | 0.877 | 0.936 | 0.939 | 0.950 | 0.954 |

## 5 Data Analysis: A Nutrition-Environment Interaction Study in Bangledash Birth Cohort

### 5.1 Study Overview

The Bangladesh reproductive cohort study was initiated in 2008 to investigate the effects of prenatal and early childhood exposure to arsenic (As), manganese (Mn), and lead (Pb) on early childhood development. During 2008-2011, pregnant female participants (with gestational age < 16 weeks) were recruited from two rural health clinics operated by the Dhaka Community Hospital Trust (DCH) in the Sirajdikhan and Pabna Sadar upazilas of Bangladesh. During 2008-2013, data are collected at five time points spanning the entire perinatal and early childhood period, including: initial clinic visit (gestational age < 16 weeks, Visit 1); pre-delivery clinic visit (gestational age = 28 weeks, Visit 2), time of delivery (Visit 3), post-delivery clinic visit (infant age less than 1 month, Visit 4), follow-up visit (infant age between 20-40 weeks, Visit 5).

*Covariates*

The detailed data collection and measurement procedures have been documented in previous literature [3, 5, 10]. Briefly, background information on parent's demographic status, including age, education, smoking history and socioeconomic status are collected through structured questionnaires at the two during-pregnancy visits to the clinic (Visit 1-2). Information on infant's biometric measurements, including sex, birth weight, length, head circumference, birth order and gestational age, are recorded at birth and extracted from medical records. Information on infant's early childhood development, including maternal and child's medical history, quality of home environment (in terms of emotional, social, and cognitive stimulation, measured by HOME instrument score [1], maternal depression status (in Edinburgh Depression scale), maternal IQ (assessed using the Raven's Progressive Indices [9] are measured at during pregnancy visits (Visit 1-2), time of delivery (Visit 3), and the follow-up visits (Visit 5), respectively.

*Exposures and Outcome*

Each infant's exposure to multiple metals As, Mn and Pb (concentrations in Ml/g) during pregnancy were measured using blood samples from infant's umbilical cord venous blood collected at the time of the birth. Mother's total nutrition intake status during pregnancy are measured for 27 nutrients derived from semi-quantitative Food Frequency Questionnaires (FFQs) specially adapted to Bangladeshi diet [6] at both the pre- and post-delivery visits (Visit 2 and 4). This instrument measures the consumption frequency (amount per week) of 42 food items during the 12-month period preceding delivery. The nutrients measured are roughly grouped into 5 categories including macro-nutrients (5 nutrients: protein, fat, carbohydrate, dietary fiber and ash), minerals (8 nutrients: calcium, iron, magnesium, phosphorus, potassium, sodium, zinc and copper), vitamin and provitamin A (6 nutrients: vitamin A, retinol, beta-carotene equivalents, alpha-carotene, beta-carotene, and cryptoxanthin), vitamin B (5 nutrients: thiamin (B1), riboflavin (B2), niacin (B3), vitamin B6 and folate (B9)), and other vitamins (3 nutrients: Vitamin C (i.e. L-ascorbic acid), Vitamin D, and Vitamin E). Finally, infant's

neurodevelopmental outcomes were assessed at 20–40 months of age (Visit 5) using a translated and culturally-adapted version of the Bayley Scales of Infant and Toddler Development, Third Edition (BSID-III) including five cognitive domains: cognitive, receptive language, expressive language, fine motor and gross motor. A growing body of literature has identified nonlinear associations between multiple metal exposures and neurodevelopment. For example, a cohort study at Mexico identified a inverted "U" relationship between manganese exposure on child development, as well as a synergistic interaction between lead and manganese [2]. Also in the Bangedash cohort, a recent study has identified U-shaped effect of manganese, and potential synergism between arsenic and manganese among the Pabna population [10]. Despite the field's growing understanding of the neurotoxic effect of prenatal exposure to metal mixture, the joint effect of all-category dietary nutrition intake on infant development, as well as the interaction effect between multi-category nutrition intake and metal mixture on infant development, remains open questions of great public health significance awaiting careful investigation.

## 5.2 Nutrition-Environment Interaction Analysis

In the current study, our aim is to detect whether mother's nutrient intake during pregnancy modifies the effect of metal mixture exposures on children's early stage neurodevelopment. We conduct a group-based analysis to study the interaction between the As, Mn, Pb mixture and five major nutrient groups: macro-nutrient, minerals, vitamin As, vitamin Bs and other vitamins. We examine the modification effect of each nutrient group on both the marginal effect of each individual pollutant, as well as the joint effect of entire metal mixture, while accounting for interaction among other covariate sets. Specifically, recall the assumed model (1):

$$y_i = \mathbf{x}_i^T \boldsymbol{\beta} + h(\mathbf{z}_i) + \epsilon_i,$$

where $\mathbf{x}_i$ is an $18 \times 1$ vector of background covariates, including child's sex, gestational age, delivery type, birth order, child's age at follow-up visit, maternal and paternal education, secondhand smoke exposure, HOME instrument scores (5 variables: Emotion, Avoidance, Caregiving, Organization, Provision, Stimulation), total energy intake during pregnancy(kcal), and child's blood concentration of As, Mn and Pb at the time of neurodevelopmental testing. $\mathbf{z}_i$ is the $30 \times 1$ vector of during-pregnancy exposure to 27 nutrients and 3 metal pollutants, corresponding the grouping structure $\mathbf{z}_i = \{\mathbf{z}_{\texttt{metal}}, \mathbf{z}_{\texttt{nutr}}\}$, where $\mathbf{z}_{\texttt{nutr}}$ is further divided into $\mathbf{z}_{\texttt{nutr}} = \{\mathbf{z}_{\texttt{macro}}, \mathbf{z}_{\texttt{mineral}}, \mathbf{z}_{\texttt{vitA}}, \mathbf{z}_{\texttt{vitB}}, \mathbf{z}_{\texttt{vitO}}\}$.

### 5.2.1 Interaction Test in the presence of Nuisance Interactions

When testing for the interaction between metal mixture exposures and a specific nutrient group of interest, care should be given to formulate $h(\mathbf{z}_i)$ such that in addition to explicitly characterizing the interaction of interest, $h$ should also account for all nuisance interactions among other $\mathbf{z}_i$ subgroups. For example, when testing for the interaction between metal exposures and the macronutrients $\mathbf{z}_{\texttt{macro}}$, the interaction between metal exposures and the other non-macro nutrients (denoting as $\mathbf{z}_{\texttt{other}}$), as well as the interaction between macro- and non-macro-nutrients, should also be included in the model. To this end, following the tensor-product construction shown in Section 5, we adopt the follwing orthogonal decomposition of $h(\mathbf{z})$:

$$\begin{aligned}
h(\mathbf{z}_{\texttt{toxin}}, \mathbf{z}_{\texttt{macro}}, \mathbf{z}_{\texttt{other}}) = &\, h_1(\mathbf{z}_{\texttt{toxin}}) + h_2(\mathbf{z}_{\texttt{macro}}) + h_3(\mathbf{z}_{\texttt{other}}) + \\
&\, h_{13}(\mathbf{z}_{\texttt{toxin}}, \mathbf{z}_{\texttt{other}}) + h_{23}(\mathbf{z}_{\texttt{macro}}, \mathbf{z}_{\texttt{other}}) + h_{12}(\mathbf{z}_{\texttt{toxin}}, \mathbf{z}_{\texttt{macro}}) + \\
&\, h_{123}(\mathbf{z}_{\texttt{toxin}}, \mathbf{z}_{\texttt{macro}}, \mathbf{z}_{\texttt{other}})
\end{aligned}$$

Under such construction, the null hypothesis of no interaction between metal exposures and macronutrient group corresponds to $h_{12}$ and $h_{123}$ equaling zero, i.e.:

$$\begin{aligned}
H_0: \quad & h = h_1 + h_2 + h_3 + h_{13} + h_{23} \\
H_a: \quad & h = h_1 + h_2 + h_3 + h_{13} + h_{23} + h_{12} + h_{123},
\end{aligned}$$

and the corresponding garrote kernel for $h \in \mathcal{H}$ is $k_\delta(\mathbf{z}, \mathbf{z}') = k_0(\mathbf{z}, \mathbf{z}') + \delta * k_a(\mathbf{z}, \mathbf{z}')$, where:

$$\begin{aligned}
k_0 &= k_1 + k_2 + k_3 + k_{13} + k_{23} \\
k_a &= k_{12} + k_{123},
\end{aligned}$$

where $k_1(\mathbf{z}_{\texttt{toxin}}, \mathbf{z}'_{\texttt{toxin}})$, $k_2(\mathbf{z}_{\texttt{macro}}, \mathbf{z}'_{\texttt{macro}})$, $k_3(\mathbf{z}_{\texttt{other}}, \mathbf{z}'_{\texttt{other}})$ are the reproducing kernels for the main-effect space of metal mixture, macronutrients, and non-macronutrients, respectively, and

similar to Section 5, we construct the higher-order interaction kernels as $\forall (i,j), k_{ij} = k_i * k_j$ and $k_{123} = k_1 * k_2 * k_3$. Consequently, denoting $\mathbf{K}_i$ as the kernel matrix corresponding to $k_i$, the null kernel matrix $\mathbf{K}_0$ and the derivative kernel matrix $\delta \mathbf{K}_0 = \mathbf{K}_a$ are

$$
\begin{aligned}
\mathbf{K}_0 \quad &= \mathbf{K}_1 + \mathbf{K}_2 + \mathbf{K}_3 + \mathbf{K}_1 \circ \mathbf{K}_2 + \mathbf{K}_2 \circ \mathbf{K}_3 \\
\delta \mathbf{K}_0 = \mathbf{K}_a &= \mathbf{K}_1 \circ \mathbf{K}_2 + \mathbf{K}_1 \circ \mathbf{K}_2 \circ \mathbf{K}_3.
\end{aligned}
$$

Following Algorithm 1, we first fit the Gaussian process model to data $(\mathbf{y}, \mathbf{X}, \mathbf{Z})$ using REML

$$
\mathbf{y} = \mathbf{X}\boldsymbol{\beta} + \mathbf{h} + \boldsymbol{\epsilon}, \qquad \text{where} \qquad \mathbf{h} \sim MVN(0, \tau\mathbf{K}_0), \quad \boldsymbol{\epsilon} \sim MVN(\mathbf{0}, \sigma^2\mathbf{I}),
$$

thereby producing model estimates $(\widehat{\boldsymbol{\beta}}, \widehat{\tau}, \widehat{\sigma}^2)$. Using the resulting estimates, we compute the score test statistic:

$$
\widehat{T}_0 = \widehat{\tau} * (\mathbf{y} - \mathbf{X}\widehat{\boldsymbol{\beta}})^T \mathbf{V}_0^{-1} \Big[ \partial \mathbf{K}_0 \Big] \mathbf{V}_0^{-1}(\mathbf{y} - \mathbf{X}\widehat{\boldsymbol{\beta}})
$$

where $\mathbf{V}_0 = \widehat{\sigma}^2\mathbf{I} + \widehat{\tau}\mathbf{K}_0$, the Satterthwaite parameters of the null distribution $(\widehat{\kappa}, \widehat{\nu})$, and also the p-value $p = P(\chi^2_{\widehat{\nu}} > \widehat{T}/\widehat{\kappa})$ as described in Section 3.

### 5.2.2 Results

We conduct the interaction test described in Section 5.2.1 to formally test for the presence of interaction between each nutrition group (as introduced in Section 5.1) and all possible combinations of the metal mixture (As, Pb, Mn) on infant's fine motor BSID-III score, which tests for infants' coordination of small muscles, e.g. hand-eye coordination. As shown in Table 10, the analysis yielded statistically significant evidence of interaction (p-value $< 0.05$) between arsenic (As) and all micronutrient groups, and suggestive evidence (p-value $< 0.1$) for the interaction between maganese (Mn) and non-B Vitamins (i.e. "Vitamin A" and "Vitamin, Other"). We also observed suggestive evidence for the interaction between (As, Mn) mixture and all micronutrient groups.

Table 10: $p - value$ for Nutrient - Environment Interaction Test

| Toxin/Nutrient | Macro | Mineral | Vitamin A | Vitamin B | Vitamin, Other |
|:---:|:---:|:---:|:---:|:---:|:---:|
| As | 0.0859 | 0.0263 | 0.0072 | 0.0357 | 0.0046 |
| Pb | 0.7291 | 0.6678 | 0.2209 | 0.4331 | 0.6586 |
| Mn | 0.1829 | 0.5482 | 0.0761 | 0.2845 | 0.0227 |
| As, Pb | 0.3835 | 0.2460 | 0.0389 | 0.0271 | 0.0078 |
| As, Mn | 0.1354 | 0.0339 | 0.0015 | 0.0469 | 0.0020 |
| Pb, Mn | 0.5810 | 0.5945 | 0.0556 | 0.2745 | 0.5381 |
| As, Pb, Mn | 0.2131 | 0.1443 | 0.0689 | 0.1248 | 0.0052 |