[Reviews · NeurIPS 2017]

Reviewer 1



The paper proposes a statistical test for particular non-linear effects in a linear mixed model (LMM). The problem of testing non-linear effects is relevant, especially in the natural sciences. The novelty is significant. The experimental validation has its flaws, but may be considered acceptable for a conference paper. The method consists of multiple parts: 1) The main new idea introduced in the paper is to introduce a kernel parameter (garotte) that interpolates between a null model and the desired alternative model and to perform a score test on this parameter. This elegant new idea is combined with several established steps to obtain the final testing procedure: 2) Defining a score statistic and deriving an approximate null distribution for the statistic based on the Satterthwaite approximation. 3) The use of cross-validation to chose appropriate kernel parameters for the null model. 4) An application, where multiplicative interactions are being tested. The title is somewhat ambiguous, as "Functional Effect" is not clear. Initially I assumed it to mean multivariate, or time-series, as in the related https://www.ncbi.nlm.nih.gov/pubmed/28056190 . Consequently, the authors may want to change the term Functional. Also, the title claims that the test is "robust". While the authors discuss the effets of kernel choice and model missmatch on robustness (power and T1 error) and propose the use of cross validation to alleviate this, the experiments suggest that robustness is not achieved in practice. The experiments rather show that the choice of the kernel has a great influence on the results. It is hard to disentangle power and T1 error in the same experiment, as done by the authors. This may yield misinterpretation of the results. Instead of arguing that simple models yield inflated T1 error, one could also argue that the Satterthwaite approximation does not yield an accurate null distribution, leading to inflated or deflated T1 error, depending on the model. A better way to assess T1 error would be a set of T1 simulations with varying alpha levels than 0.05. Also, from an applied perspective it should be argued that the linear model does not have inflated T1 error, but rather does exactly what it is supposed to do, namely detect multiplicative effects in the data. Given the kernel functions used (Matern and RBF) the functions involve multiplicative effects in the input variables, even under the null simulation. These issues should at least be discussed.

Reviewer 2



The authors provide a mechanism for establishing hypothesis tests of functional forms based on Gaussian process descriptions. First, following [12] the Gaussian process predictor is represented as a linear mixed effects model. Second, using [11], the authors use the classical variance component test to develop a testing procedure for hypothesis involving GPs. Third, following [13], the kernel function is replaced by a garrote kernel function which simplifies the application of the test. The test is illustrated with a simulated data example. I found the reading of the paper extremely involved. There is a lack of explanations of the basic methods, and the reader needs to refer constantly to [11], [12] and [13] to follow the general ideas of the paper. It would be nice if the paper could be made self-contained or at least, the key ideas provided in a supplemental material. I have two criticisms: (1) why doing this is more convenient/better than doing model selection for GPs, i.e. using ARD priors to select which features are more relevant? (2) how could a practitioner use the test? The application of the test could be better described so that it could actually be used. Examples with real data can also help to better understand the method. My feeling is as the paper stands, it can hardly have an impact on the NIPS community.

Reviewer 3



This paper considers an interesting issue of hypothesis testing with complex black models. I will say from the outset that I am skeptical of the NHST framework, especially as it is currently applied in classical statistical settings, where I think it is clear the ongoing harm that it has done to applied science (see, e.g. the ASA's statement on p-values). Whether I should be more or less skeptical of it in nonparametric settings is an interesting questions and I'd be happy to be convinced otherwise. I guess the standard arguments aren't worth rehashing in great detail but just so you're aware of where I'm coming from, if I care about the (possibly non-linear) interaction effect then I want a kernel parameterized as: beta1 * k1a(x1,x1') + beta2 * k2c(x2,x2') + beta12 * k1b(x1,x1')*k2b(x2,x2') I want to fit that model using all of my expensive data, and then say something about beta12. Bayesian methods would give me a posterior credible interval for beta12---given limited data this is going to be my approach for sure and I'm going to spend time constructing priors based on previous studies and expert knowledge. But if you want to take a frequentist approach that's fine, you can just bootstrap to obtain a confidence interval for beta12, which I'm going to be happier with than the asymptotics-based approach you take for small samples. OK, now that I've gotten that disclaimer out of the way (apologies!), taking your paper on its own terms, here are some areas that I think could be improved. On line 29 I'm not sure why the null hypothesis is additive; shouldn't this be a multiplicative interaction between air pollutants and nutrient intake? Be careful in Section 2: the fact that h ~ GP with covariance kernel k means that the probability that h is in the RKHS H_k is 0! This is a surprising fact (see Wahba 1990 and Lukic and Beder 2001) and I don't think it invalidates your approach (e.g. E[h] is in the RKHS), but it's worth fixing this point. I like the GP-LMM connection and think it needs more explanation as I am not very familiar with REML. Could I, for example, posit the following model: y = h1 + h2 + epsilon, h1 ~ GP(0,K1) and h2 ~ GP(0,K2) and then use your approach to decide between these models: y = h1 + epsilon y = h2 + epsilon y = h1 + h2 + epsilon? I think the answer is yes, and I'm wondering if it'd be clearer to explain things this way before introducing the garrote parameter delta. Turning the test statistic into a model residual also helped me think about what was going on, so maybe it'd be good to move that earlier than Section 4. It also might be nice to show some training/testing curves (to illustrate bias/variance) in Section 4 to explain what's going on. It wasn't obvious to me why the additive main effects domain H1 \osum H12 is orthogonal to the product domain H1 \otimes H2 (section 5). For the experiments, if this were real data I could imagine trying all of the difference kernel parameterizations and choosing the one that the fits the best (based on marginal likelihood I guess) as the model for the null hypothesis. Of the different models you considered, which one fit the data best? And did this choice end up giving good type 1 and type 2 error? A real data experiment would be nice.